🔓 | **Open Peer Review** | Environmental Microbiology | Research Article

# Extreme environments simplify reassembly of communities of arbuscular mycorrhizal fungi

Nataša Šibanc,[1,2] Dave R. Clark,[3,4] Thorunn Helgason,[5,6] Alex J. Dumbrell,[3] Irena Maček[7]

**ABSTRACT** The ecological impacts of long-term (press) disturbance on mechanisms regulating the relative abundance (i.e., commonness or rarity) and temporal dynamics of species within a community remain largely unknown. This is particularly true for the functionally important arbuscular mycorrhizal (AM) fungi; obligate plant-root endosymbionts that colonize more than two-thirds of terrestrial plant species. Here, we use high-resolution amplicon sequencing to examine how AM fungal communities in a specific extreme ecosystem—mofettes or natural $CO_2$ springs caused by geological $CO_2$ exhalations—are affected by long-term stress. We found that in mofettes, specific and temporally stable communities form as a subset of the local metacommunity. These communities are less diverse and dominated by adapted, "stress tolerant" taxa. Those taxa are rare in control locations and more benign environments worldwide, but show a stable temporal pattern in the extreme sites, consistently dominating the communities in grassland mofettes. This pattern of lower diversity and high dominance of specific taxa has been confirmed as relatively stable over several sampling years and is independently observed across multiple geographic locations (mofettes in different countries). This study implies that the response of soil microbial community composition to long-term stress is relatively predictable, which can also reflect the community response to other anthropogenic stressors (e.g., heavy metal pollution or land use change). Moreover, as AM fungi are functionally differentiated, with different taxa providing different benefits to host plants, changes in community structure in response to long-term environmental change have the potential to impact terrestrial plant communities and their productivity.

**IMPORTANCE** Arbuscular mycorrhizal (AM) fungi form symbiotic relationships with more than two-thirds of plant species. In return for using plant carbon as their sole energy source, AM fungi improve plant mineral supply, water balance, and protection against pathogens. This work demonstrates the importance of long-term experiments to understand the effects of long-term environmental change and long-term disturbance on terrestrial ecosystems. We demonstrated a consistent response of the AM fungal community to a long-term stress, with lower diversity and a less variable AM fungal community over time under stress conditions compared to the surrounding controls. We have also identified, for the first time, a suite of AM fungal taxa that are consistently observed across broad geographic scales in stressed and anthropogenically heavily influenced ecosystems. This is critical because global environmental change in terrestrial ecosystems requires an integrative approach that considers both above- and below-ground changes and examines patterns over a longer geographic and temporal scale, rather than just single sampling events.

**KEYWORDS** arbuscular mycorrhiza, elevated $CO_2$, long-term experiments, soil biodiversity, soil hypoxia, next-generation sequencing (NGS)

Address correspondence to Irena Maček, irena.macek@bf.uni-lj.si, or Alex J. Dumbrell, adumb@essex.ac.uk.

The authors declare no conflict of interest.

See the funding table on p. 19.

E cologists have long recognized the need to understand the mechanisms regulating the relative abundance (i.e., commonness or rarity) of different species within a community (1); as it is these differences in relative abundance, and associated changes in species interactions, that promote different ecosystem functions and thus ecosystem multifunctionality (2). In recent years, attention has turned toward microbial species, which underpin most major ecosystem functions and biogeochemical processes (3, 4). Aided by advances in DNA sequencing technologies, this has led to renewed interest in community ecology and the underlying mechanisms regulating natural communities. However, despite this increased research effort, the dynamics of many functionally important microbial communities still remain poorly understood.

The arbuscular mycorrhizal (AM) fungi are one of the most important microbial groups in terrestrial ecosystems. They are obligate plant-root endosymbionts that colonize approximately two-thirds of terrestrial plant species (5). In return for using plant photosynthates as their sole carbon source, AM fungi alter their hosts' perform-ances via, among other things, increased phosphorus and nitrogen uptake, improved water relations and protection against pathogens (6–9). However, whether this symbiosis provides a net benefit to the host and what the magnitude of this benefit is, also depends to some extent on the identity of both the AM fungi and host-plant species, with some plant-AM fungal interactions providing greater benefits than others (10, 11). Thus, any changes in the relative abundance of AM fungal species can change the competition dynamics within that plant community (10). Consequently, the abundance of plant species within an ecosystem is determined partially by which AM fungal species are locally abundant (12).

Our understanding of the mechanisms regulating AM fungal communities has increased rapidly in recent years. AM fungi cannot be identified morphologically in roots (13) and to overcome these problems, DNA-based techniques were developed to examine AM fungi in field-collected plant roots [e.g., references (14–18)], with next- (second or third) generation sequencing technologies now routinely employed in this quantification (19, 20). A consensus is emerging that the composition of AM fungal communities is primarily determined by an environmental filter reflecting local soil physiochemical properties (21–23) and anthropogenic disturbance (24), and that host-plant identity is of secondary importance in structuring these communities (25). Within the local species pool, the relative abundance of AM fungi and the identity of dominant species appear to be largely regulated by priority effects (15, 26). This produces a highly consistent, yet idiosyncratic property of AM fungal communities— pronounced overdominance of the most abundant taxa—which consistently comprises over 40% of the relative abundance of the community (15, 27). These patterns of dominance are outside those predicted by any standard community-assembly model (15). Moreover, these patterns do not reflect a single and ubiquitous dominant taxon as the dominant taxon's identity is different across studies (15) and unlikely to be the result of any methodological biases [see reference (18)].

Dumbrell et al. (15) proposed an explanatory mechanism for this observation based on priority effects in temperate grassland systems. At the start of the main growth period, the AM fungal taxon which colonizes a plant root first, will gain a disproportion-ate share of plant carbon enabling greater growth and further proliferation. This leads to a positive feedback effect, which promotes overdominance by a single taxon, but at the end of the growth period the system resets. During the next growth period, the identity of the dominant taxon may change and is largely controlled by a stochastic selection process (15, 28). While this theory proposes a parsimonious mechanism behind observed patterns of relative abundance of AM fungal taxa within a community, it remains largely experimentally untested, although observationally well supported [e.g., references (15, 23, 26, 27)].

These proposed mechanisms for structuring AM fungal communities are hierarchical. First, an environmental filter based around local soil conditions selects for the local pool of AM fungal taxa that are drawn from the wider metacommunity. From these,

the identity of the dominant taxon is drawn largely at random (a stochastic process) with priority effects regulating that taxon's abundance. However, a stronger or extreme environmental filter could overturn these processes, by selecting specific locally adapted types leading to competitive exclusion of other AM fungal taxa as indicated in our previous study on the AM fungal community (29). The mechanism behind this has yet to be fully examined and would require a model system that has a stable, long-term, and extreme environmental filter, from which the AM fungal community is tracked temporally. One potential model system is mofettes, where carbon dioxide ($CO_2$) of ambient temperature and geological origin reaches the soil surface, resulting in localized hypoxia and soil acidification over small spatial scales (29–32). Mofettes represent a stable long-term gradient of the local soil abiotic environment and because $CO_2$ is geological in origin and vents through the soil, plant roots and soil organisms are the first to be affected by this $CO_2$ source (29, 33–35). In these mofette environments AM fungal communities are likely to be dominated by a hypoxia-tolerant specialist (29), and this pattern is likely to be more temporally stable.

In this study, we use 454-pyrosequencing to examine the AM fungal community from a well-characterized mofette system, and test the following hypotheses, based on our previous findings (29): (H1) environmental extremes in mofettes limit diversity; (H2) in environments with long-term abiotic extremes, a distinct and consistent community of AM fungi will exist, which is a nested subset of the larger local metacommunity; (H3) environmental extremes will have a stabilizing effect on AM fungal communities removing or reducing the influence of stochastic processes and priority effects in determining the identity of the dominant taxon. By tracking this system across seasons and over 2 years, with additional comparison with the sequences from our previous study in mofettes (29), we also examine whether (H4) yearly fluctuations in dominant taxa identity, and within-season fluctuations in dominant taxa's abundance will be removed in extreme environments but present in controls. Finally, we expand this study to incorporate additional mofette sites across Europe and examine whether (H5) the influence of abiotic extremes is independent of habitat or geographical context.

## RESULTS

We analyzed 36,208 sequences across all four study sites out of a total of 62,733 raw sequences once reads that did not meet our quality control criteria were removed (see Materials and Methods section for more details). These sequences comprised a total of 37 MOTUs (EMBL accession numbers: LK937169 to LK937200; and GenBank accession numbers: MG461856 to MG461860; Fig. S1) of AM fungi from 54 samples (Slovenian meadow, $n = 36$; Slovenian forest, $n = 10$; Czech site, $n = 2$; and Italian site, $n = 6$. Rarefaction curves showed that, after communities were rarefied to an even depth (229 reads per sample), communities were sufficiently deeply sequenced to capture most AM fungal diversity (Fig. S4).

### AM fungal diversity nestedness and community composition of the mofette sites

We found statistically clear differences in AM fungal diversity between high $CO_2$ exposed and control communities in the Slovenian meadow site for all three of the diversity metrics analyzed (Fig. 1A, $H_0$; coef = −0.71, $z = -4.88$, $P < 0.001$, $H_2$; coef = −1.73, $t = -4.46$, $P < 0.001$, $H\infty$; coef = −0.89, $t = -3.57$, $P < 0.001$). High $CO_2$ exposed communities contained approximately half the number of species and were more strongly dominated by the most abundant species than control communities (relative abundance of the dominant MOTU; high $CO_2$ sites = 61.9%; control sites = 34.9%).

NMDS analysis of the AM fungal communities showed that high $CO_2$ exposed Slovenian meadow site communities were distinct from those in control samples (Fig. 1B). PERMANOVA analysis of Sorensen pairwise dissimilarities between communities further confirmed that communities were more dissimilar between $CO_2$ conditions than within ($F = 13.42$, $R^2 = 0.28$, $P < 0.001$). To determine whether high $CO_2$ exposed

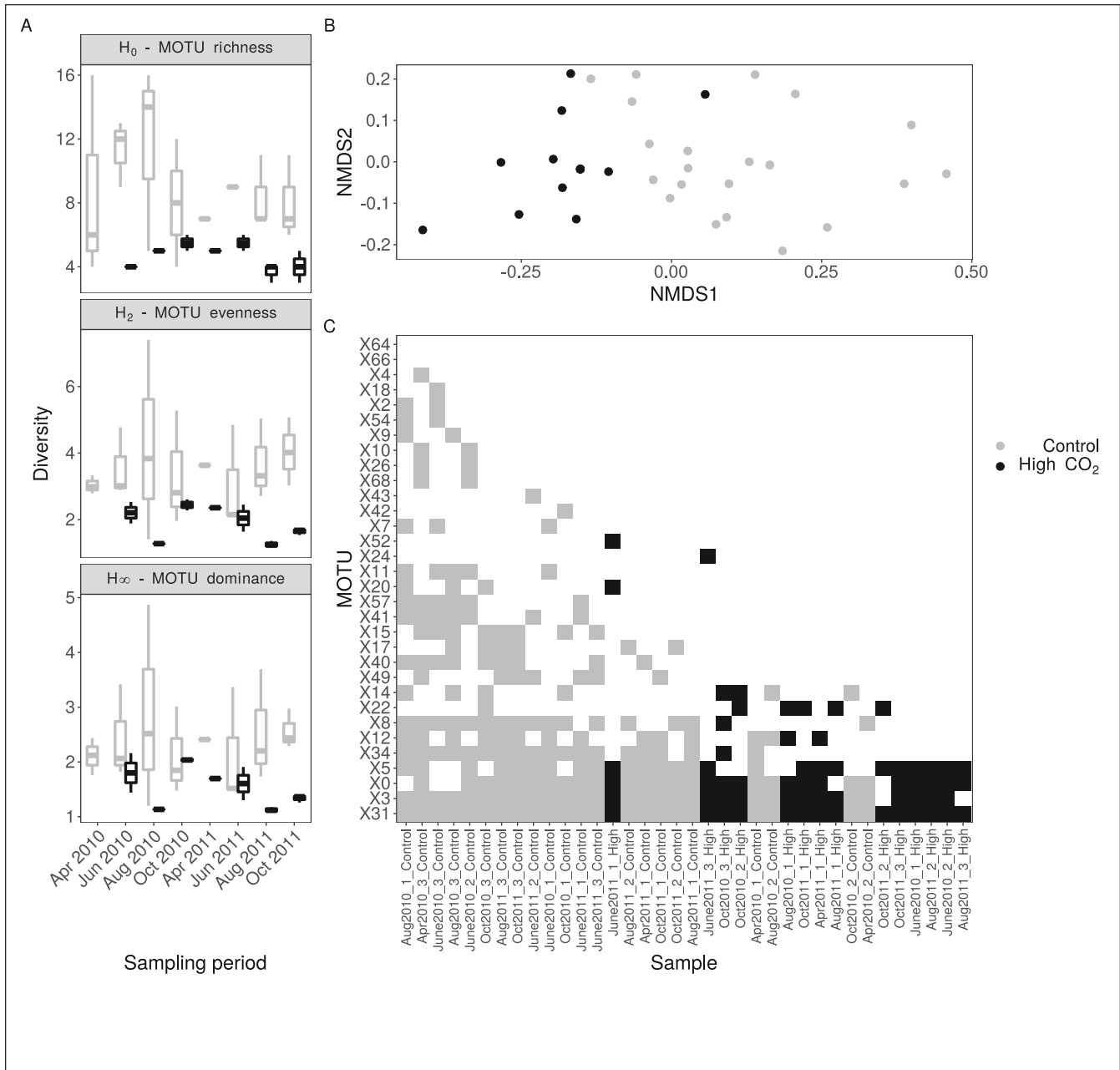

**FIG 1** The diversity, community composition, and nestedness of AM fungal communities from the Slovenian meadow site. (A) Diversity was quantified as MOTU richness (H0), evenness (H2), and dominance (H∞). (B) An NMDS (non-metric multidimensional scaling) analysis of AM fungal communities, based on a Sorensen dissimilarity matrix. Stress was calculated as 0.12. (C) An optimally packed MOTU presence-absence matrix from which matrix temperature was calculated.

communities represented a nested subset of the metacommunity, or contained distinct (potentially locally adapted) species, we partitioned pairwise dissimilarity into a nested-ness and turnover component. Additional PERMANOVA analyses of these nestedness and turnover components showed statistically clear effects of $CO_2$ in both cases, although the effect was notably stronger for nestedness than for turnover ($\beta_{nes}$; pseudo-$F$ = 24.90, $R^2$ = 0.42, $P$ < 0.001, $\beta_{sim}$; pseudo-$F$ = 5.98, $R^2$ = 0.15, $P$ < 0.001), supporting our hypothe-sis that the high $CO_2$ exposed communities are nested subsets of the local metacom-munity. The matrix temperature of our optimally packed community matrix (sites sorted by species richness, species sorted by occupancy) averaged 14.2 (std. error = 0.002), suggesting that communities showed a strong degree of nestedness. Inspection of the

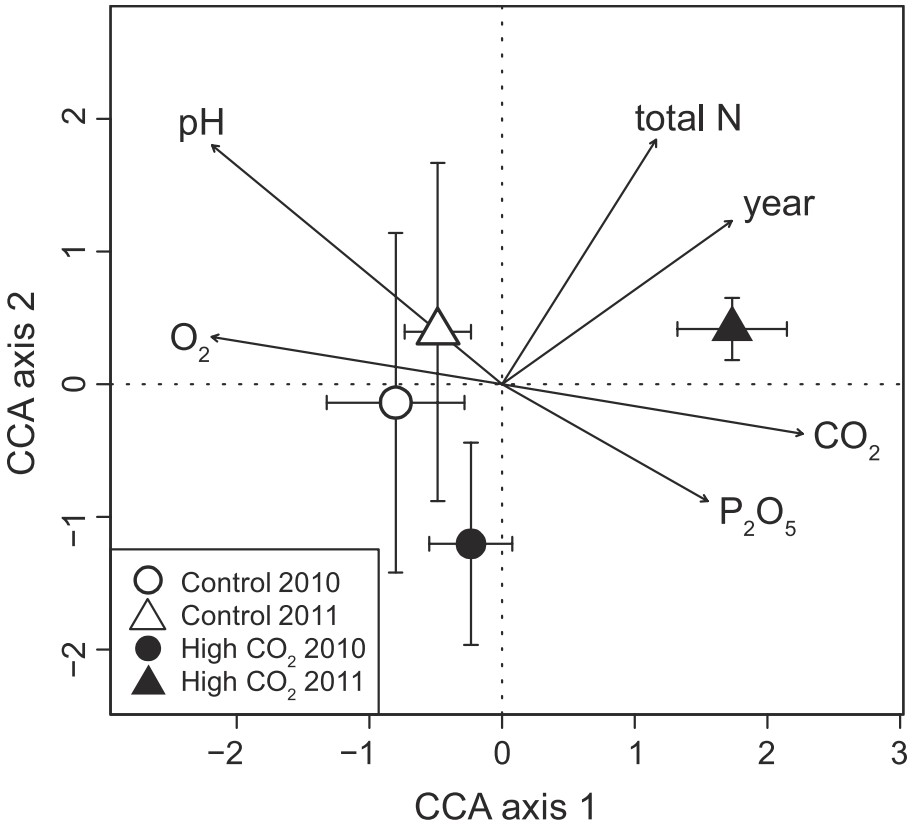

**FIG 2** AM fungal community CCA (canonical correspondence analysis) plot for the Slovenian meadow site. The CCA axis 1 explained 43.66% of AM fungal variability in AM fungal community composition, and CCA axis 2 21.37% of variability. In the CCA plot, samples are pooled by month, and points represent mean ± standard deviation of CCA scores for each axis. Significant ($P < 0.001$) environmental vectors are presented on CCA plot.

position of high $CO_2$ exposed communities in this matrix revealed that these communities tended to be toward the right of the matrix, and therefore are subsets of control communities (Fig. 1C).

The differences between the Slovenian meadow site high $CO_2$ exposed and control AM fungal community were also visualised using canonical correspondence analysis (CCA) which explained 65.03% of the variation within the taxa abundance data across the first two ordination axes (Fig. 2; Fig. S2). Permutation test of CCA under a reduced model (10,000 permutations) showed significant association between AM fungal composition at Slovenian meadow sites and selected environmental variables ($P < 0.005$). AM fungal assemblages, exposed to high $CO_2$ were significantly separated in the PERMANOVA from the controls for the environmental vectors $CO_2$ ($R^2 = 0.63$) and $P_2O_5$ ($R^2 = 0.38$) (Fig. 2; Fig. S2). Control communities were separated from high $CO_2$ exposed AM fungal communities in the direction of environmental vectors pH ($R^2 = 0.96$) and $O_2$ concentration ($R^2 = 0.59$) (Fig. 2; Fig. S2). While the difference between 2010 and 2011 sampling was in the direction of total N ($R^2 = 0.57$) and sampling years ($R^2 = 0.54$). None of the weather variables were significant drivers of AM fungal composition, nor was the sampling month.

## Stability of high $CO_2$ exposed mofette AM fungal community and annual fluctuations in dominant taxa identity

The stability of AM fungal communities sampled from Stavešinci meadow mofette areas was first assessed as the presence or absence of the most abundant MOTUs in high $CO_2$ exposed areas from 2007 until 2011. MOTU5 and MOTU22 remain present in high $CO_2$

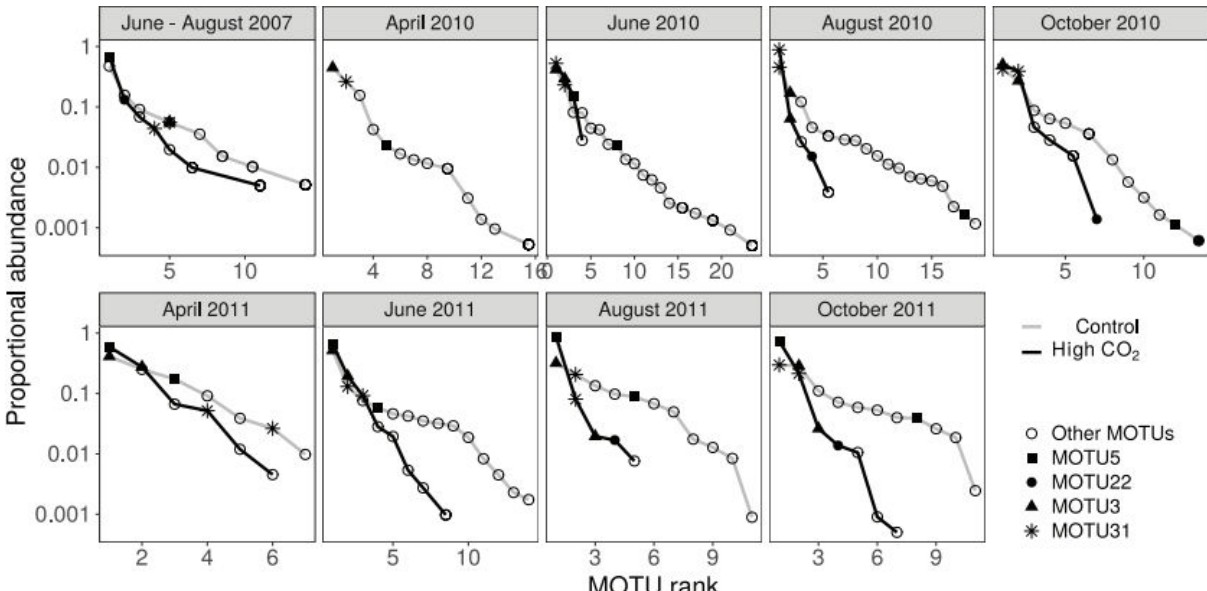

**FIG 3** Unnormalised rank abundance of MOTU5 and MOTU22, most abundant in mofette areas with high $CO_2$ exposure from this paper, and MOTU31 and MOTU3, most abundant in control areas from this paper presented for sampling in June–August 2007 (29) and from April 2010 till October 2011, separately for each sampling time within the Slovenian meadow sites.

exposed mofette areas (Fig. 3), but the rank position shifts in between high $CO_2$ exposed and control areas and among years 2007 (24), 2010, and 2011 (Fig. 3). MOTU5 was the most abundant MOTU in high $CO_2$ exposed area in 2007 sampling, and throughout sampling in 2011 (Fig. 3) and, like MOTU22, aligned most closely with virtual taxon VT309 (Fig. S1) mostly found in anthropogenically disturbed sites (e.g., metal polluted sites and agricultural sites), according to our phylogenetic analyses.

Likelihood ratio tests of our MV-GLMs showed that for 10 of the 18 modeled AM fungal MOTUs, $CO_2$ had a statistically clear effect ($P < 0.04$ for all cases). The two most abundant MOTUs in high $CO_2$ exposed conditions (MOTUs 5 and 22) both showed statistically clear effects of $CO_2$ (MOTU5; $CO_2$ coefficient = 4.73, deviance = 17.32, $P < 0.001$, MOTU22; $CO_2$ coefficient = 18.66, deviance = 8.21, $P < 0.04$). In contrast, there was no statistically clear effect of $CO_2$ on the most abundant MOTU under control conditions (MOTU3; $CO_2$ coefficient = −3.15, deviance = 4.49, $P = 0.14$).

According to our hypothesis 4, we expected the MOTUs that were more abundant under high $CO_2$ exposure to show seasonally stable abundances. However, MOTU5 showed statistically clear differences in abundance between sampling periods (deviance = 53.19, $P < 0.01$). In contrast, MOTU22 showed no statistically clear differences in abundance between sampling periods (deviance = 7.58, $P = 0.83$).

**TABLE 1** Results of (generalized) linear mixed effects models, relating three Hill measures of diversity to $CO_2$ (high $CO_2$ exposed vs control)[a]

| Model | $CO_2$ coefficient | Test statistic | P value |
|---|---|---|---|
| $H_0$—all data | −0.33 (0.19) | −1.73 | 0.08 |
| $H_0$—excluding Slovenian forest communities | −0.58 (0.13) | −4.61 | <0.001 |
| $H_2$—all data | −1.01 (0.75) | −1.34 | 0.18 |
| $H_2$—excluding Slovenian forest communities | −1.60 (0.36) | −4.42 | <0.001 |
| $H∞$—all data | −0.54 (0.39) | −1.39 | 0.16 |
| $H∞$—excluding Slovenian forest communities | −0.81 (0.22) | −3.72 | <0.001 |

[a]Model represents the Hill number being modeled, and the data being used. $CO_2$ coefficients represent change in diversity compared to control group, standard errors of coefficient estimates are given in parentheses. All statistics are reported to two decimal places.

## Habitat and geographical context of AM fungal diversity and community composition

In addition to the Slovenian meadow sites, we analyzed AM fungal communities from Slovenian forest, Italian and Czech sites to examine whether the effects of elevated $CO_2$ are dependent on site or habitat (hypothesis 5). Our models indicated that $CO_2$ did not have a statistically clear "global" effect on AM fungal diversity across study sites (Table 1). However, for all three diversity metrics, when we removed forest sites from the data set, we once again found that elevated $CO_2$ reduced diversity, suggesting potential interactions with habitat type (Fig. 4; Table 1).

Multivariate modeling of the AM fungal communities across all four sites yielded three MOTUs for which $CO_2$ had a statistically clear effect (MOTUs 5, 22, and 34; deviance >9.37, $P < 0.04$ in all cases), whereas five MOTUs showed clear site effects (MOTUs 0, 6, 12, 23, and 31, deviance >14.99, $P < 0.02$ in all cases). Of the MOTUs that showed $CO_2$ effects, only MOTU5 was present across all four study sites, where it was on average 2.3-fold more abundant in elevated $CO_2$ conditions compared to ambient. MOTU22 was only present in the Slovenian meadow site, indicating that high $CO_2$ communities comprise a mix of species with varying biogeographic distributions.

We can observe distinct AM fungal communities from mofette areas, in the direction of $CO_2$ environmental vector in the CCA (Fig. 5). However, this was not the case for Italian sites (Fig. 5). Among the significant environmental vectors, the most variability in AM fungal composition was explained by pH ($R^2 = 0.85$), followed by $CO_2$ concentration ($R^2 = 0.73$), total N ($R^2 = 0.70$), and concentration of $O_2$ ($R^2 = 0.65$) (Fig. 5; Fig. S3).

The most abundant MOTU5 (assigned to the MaarjAM database VT309) from high $CO_2$ exposed areas in 2007 sampling, and throughout sampling in 2011 from Slovenian meadow sites (Fig. 3) was also found in all other sites (Italy, Czech, and in Slovenian forest site). It was the most abundant MOTU from high $CO_2$ exposed areas within Czech mofettes, second most abundant from Slovenian forest sites and third most abundant in Italian sites (Fig. 6).

## DISCUSSION

This study reveals that long-term stress results in consistently distinct communities of AM fungi in the hypoxic soils around natural $CO_2$ springs (mofettes), forming a relatively predictable and temporally stable (consistent community across years) pattern in overall AM fungal community composition in the extreme sites over several years. Our results suggest that direct environmental selection, namely soil gas composition and consequent change in soil pH, are the main drivers structuring AM fungal commun-

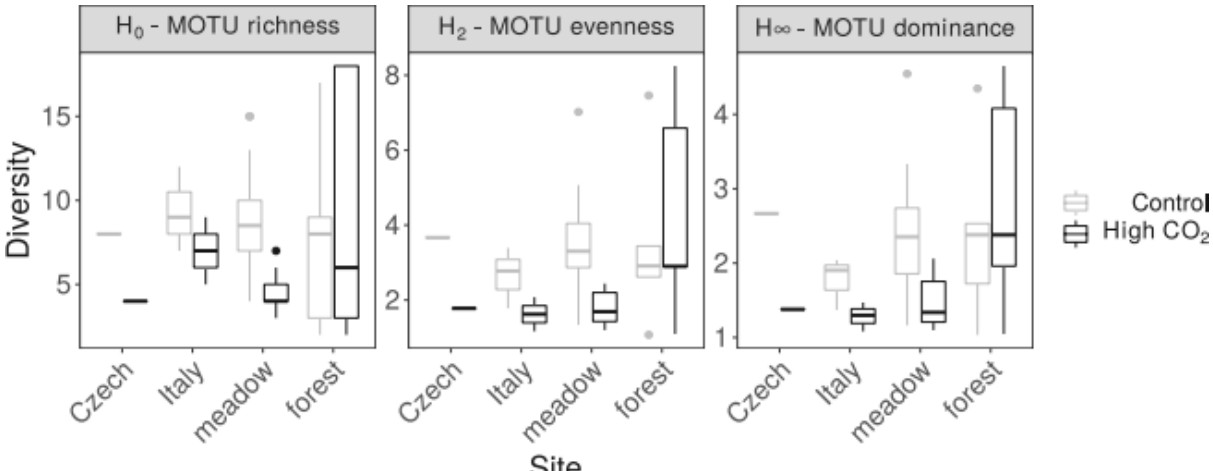

**FIG 4** The diversity of AM fungal communities from the sampled mofette areas (Slovenian meadow and forest site, Czech and Italian site). Diversity was quantified as MOTU richness (H0), evenness (H2), and dominance (H∞).

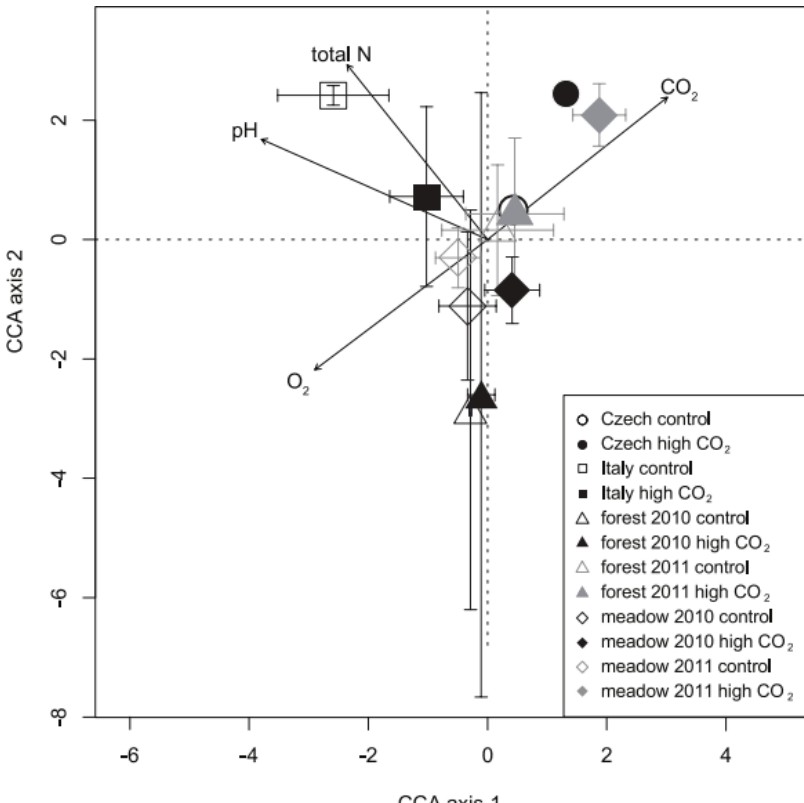

**FIG 5** AM fungal community CCA (canonical correspondence analysis) plot for Slovenian meadow and forest sites, Italian and Czech sites. The CCA axis 1 explained 57.36% of AM fungal variability in AM fungal community composition, and CCA axis 2 21.50% of variability. In CCA plot, samples are pooled, and points are representing mean ± standard deviation of CCA scores for each axis. Significant ($P < 0.001$) environmental vectors are presented on CCA plot.

ities across sampled mofette areas and that the influence of these abiotic extremes is independent of geographical context (mofette sites in Slovenia, Italy, and Czechia), despite the fact that we could not find differences for the habitats (grassland vs forest), sampled only in Slovenian mofettes. The data from the Slovenian Stavešinci meadow mofettes are consistent with our previous findings on AM fungal communities in mofette environments for this site (29). The methodology of the current study, however, enables a higher sampling depth of the molecular data and therefore a more detailed insight into the mechanisms regulating the diversity of AM fungal communities in this ecosystem. Identifying the main predictors of community-level processes during a longer time scale of seasonal sampling throughout two consecutive years—2010 and 2011—and extending the data of the current study with the data set sampled in 2007 (29) within the same mofette field, increases our temporal range giving important new insights into the stability of AM fungal communities across years (2007, 2010, and 2011) and after exposure to long-term stress. These data show that communities exposed to high geological $CO_2$, with taxa abundant in the local extreme environment, represent nested subsets of the more diverse communities in control sites. The same dominant high-$CO_2$ selected taxa (both MOTU 5 and 22 fall into the same VT according to MaarjAM database) were recorded in other environments, impacted by anthropogenic stressors, which include vineyards (36, 37), heavy metal-contaminated soil (38), and monocultures of different crops like maize and rice (39, 40), while being very rare or absent in benign environments (data according to MaarjAM database). We predict that this reflects ecologically relevant functional traits of these taxa like hypoxia tolerance and intraradical life strategy (formation of more structures inside roots and in rhizosphere

with more oxygenated environment), but, development of robust techniques to study the functional traits of fungi is required to successfully characterize the ecological roles of the fungi in mofettes.

## AM fungal diversity and community nestedness in the Slovenian meadow mofette site

Slovenian meadow mofette communities in roots of plants exposed to high $CO_2$ soils were consistently distinct from those in control habitats and contained approximately half the number of taxa (Fig. 1). The high $CO_2$ exposed communities were less even, and more strongly dominated by the most abundant taxa than control communities (Fig. 1). Lower richness in high $CO_2$ exposed sites is associated with the higher abundance of the dominant taxon and is consistently observed in AM fungal community ecology studies (15, 24, 27), along with typical patterns of dominance, where 76% of sequences belonged to three dominant MOTUs (MOTU3, MOTU31, and MOTU5). Extremes limit diversity, and thus the pool of species that may become dominant is much smaller and therefore more temporally stable. In the only published study on AM fungal diversity in mofette areas, Maček et al. (29) reported a numerical dominance by two AM fungal phylotypes in hypoxic soils. The study suggested that community assemblages of AM fungi are the result of powerful selection by, and local adaptation to, the soil environment. As predicted by the authors (29), more intensive sampling and different sequencing technologies used in the present study, have revealed these phylotypes to be rare taxa within the wider AM fungal community and not to be completely absent in the control community. Longer‐term studies, including natural long-term experiments like mofettes [see the concept in reference (31)] allow greater recruitment from the meta‐community of taxa pre‐adapted to these extreme environmental conditions and if sufficiently long-term, the evolution of new species (41), which may be tested

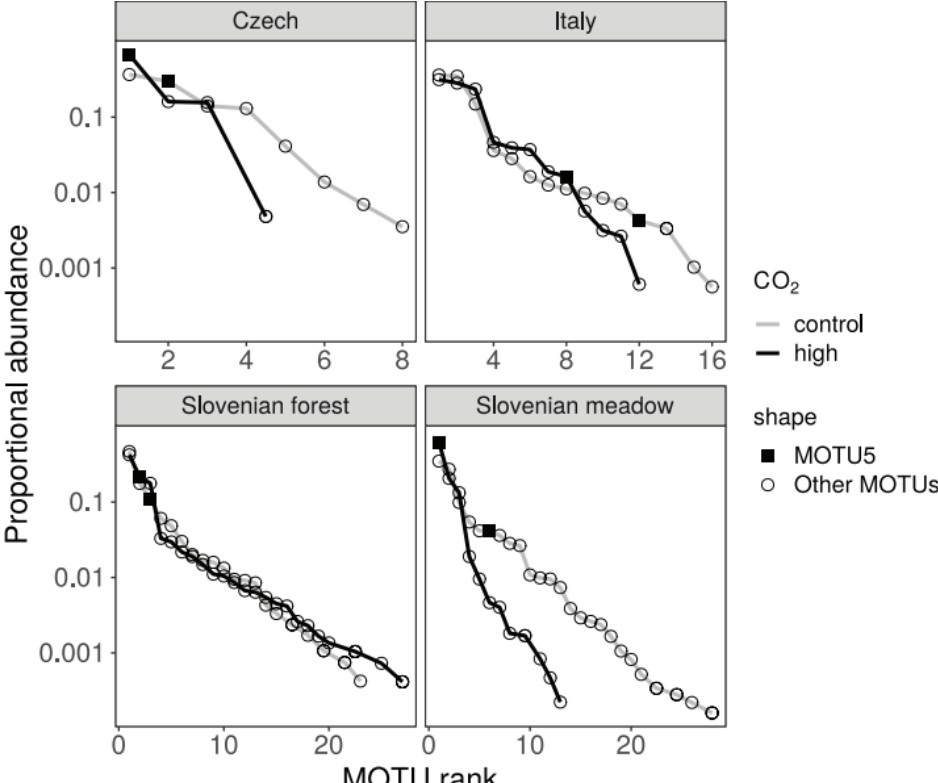

**FIG 6** Unnormalized rank abundance of the most abundant MOTU5 (assigned to the MaarjAM VT309) from mofette areas with high $CO_2$ exposure, that was also significantly more abundant at high $CO_2$ exposed areas compared to control areas.

in the future by genomic and functional tools. At the Slovenian meadow mofettes, the expansive semi‑natural grassland surrounding $CO_2$ vents and the long-term nature of the mofette site (31) certainly provide ample opportunity for recruitment of pre‑adaptive AM fungal taxa from the meta‑community.

The mofette communities are clearly nested subsets of the local control metacommunity, supporting hypothesis 2. A smaller taxon pool has been found in high $CO_2$ exposed sites as well as an almost complete shift in dominant taxa (Fig. 1). Lower diversity communities generally form subsets of taxonomically richer communities, indicating a general trend of local taxa disappearance from the meta-community (potential taxa pool) taxa to be "replaced" by supposedly already dominant taxa, rather than by alternative subordinates (27). Maček et al. (29), studying the impact of elevated $CO_2$ at the same mofette site, but using lower resolution of sequencing (*in vivo* cloning and Sanger sequencing, using 403 sequences) reported a high level of β-diversity and a complete turnover of the high $CO_2$ exposed AM fungal community, whereas in this consecutive study of the same Slovenian meadow site using deeper sequencing resolution (24,857 sequences), the AM fungal community nestedness was clearly confirmed. Designed paired sampling from three areas within the Stavešinci meadow site that were 14–46 m apart ensured that spatial isolation did not influence the AM fungal communities present in the paired high $CO_2$ and control sites. The distance between pairs (high $CO_2$ exposed locations and control) 4–14 m ensured that each of the three pairs was independent of the others. Community nestedness can be caused by (i) environmental filtering, (ii) community size, and (iii) spatial isolation (27). At mofettes, there is a clear environmental filtering caused by increased $CO_2$ and consequently lowered $O_2$ and pH and because AM fungal taxa differ in their tolerance to the environmental conditions imposed, sensitive species taxa may disappear along a gradient causing community nestedness (27). Similar effects have been found to cause nestedness of communities in response to varying levels of environmental "harshness" [e.g., pond communities in response to drought (42); and floral composition in response to elevation (43)].

## Drivers and stability of AM fungal community composition in the mofette sites

The composition of AM fungal communities was significantly different in high $CO_2$ exposed sites in the most extensively sampled Slovenian meadow mofette (Fig. 2). These findings support our initial hypotheses 2 and 3 and result from earlier work on mofette systems (29). The results strongly suggest that direct environmental selection acting on AM fungi is a major factor regulating AM fungal communities and their phylogeographic patterns in extreme conditions like mofettes (29). Consequently, some AM fungi were more strongly associated with local variations in the soil environment than with their host plant's distribution, and this was confirmed within the present study (Fig. 1 to 3) and supported by previous findings (29). Further research into the Stavešinci meadow mofette site shows that localized, long-term abiotic selection pressure does drive consistent and robust compositional differences in AM fungi and other soil microbes (35, 44–47. However, not only do soil $CO_2$ concentrations directly impact soil microbes but so do co-varying factors such as changed soil pH and local hypoxia, both of which strongly correlated with increased $CO_2$ (Fig. 2) (31). In mofette systems, $CO_2$ converts to carbonic acid in contact with soil water and consequently decreases pH (correlation between $CO_2$ and pH; $r = -0.58$; $P < 0.001$), thus making soil pH a long-term indicator of increased $CO_2$ concentrations (31). Soil pH is a good predictor of soil microbial communities in general, relevant also at larger spatial scales and at relatively coarse levels of taxonomic resolution (22, 48). The importance of soil pH in structuring AM fungal communities in natural environments was first reported in the study of Dumbrell et al. (21), where it was shown to be the main variable influencing AM fungal community composition along a pH gradient ranging from pH 3.7 to 8.0. In a global study, temperature and pH have been confirmed to be the most important abiotic drivers in AM fungal taxa distribution

along with spatial variables, the latter generally occurring at local to regional scales (22). Also, soil $O_2$ concentration is highly negatively correlated with soil $CO_2$ ($r = -0.87$; $P < 0.001$) in mofettes (31). As $O_2$ level is a major abiotic factor affecting all living biota, it was previously hypothesized that this and not direct exposure to high $CO_2$ levels is the primary determinant of AM community composition from mofette sites (29, 31). $O_2$ concentrations from control $CO_2$ exposed soils allow aerobic respiration in these areas, whereas the high $CO_2$ exposed soils are consistently hypoxic and often anoxic (29–33). Nevertheless, like their plant hosts, AM fungi are known to be aerobic organisms that need respiratory energy to support their growth and function, thus their exposure to severe hypoxia that occurs in high $CO_2$ exposed mofette sites produces a physiological response and drives community composition toward taxa potentially adapted to, or tolerant of, low $O_2$ environments (29, 31). Alternatively, the continued temporal turnover of roots is providing a dynamic source of habitats for the AM fungi (31). This would allow rarer and potentially weaker competitors a chance to colonize roots in the absence of other AM fungi. While there is evidence for this within the mofette-AM fungal data, finer-scale temporal sampling is required to demonstrate this conclusively.

High $CO_2$ exposed communities sampled in the Slovenian meadow mofette are relatively predictable in their composition with reasonably stable temporal patterns (year-to year), consistently dominated by taxa that have been confirmed previously to be more abundant in high $CO_2$ within both years and across sampled years since 2007 (29) (Fig. 3 and 6). In all samples collected, the composition of AM fungal communities significantly differed in response to high geological $CO_2$ exposure, revealed as a large re-sorting of the AM fungal community. This is in contrast to the control communities that have changed less predictably among the sampling years, as observed in other benign environments in temperate regions. In temperate European grasslands, during the main plant growth period (April to October), AM fungal communities vary little in composition (28, 49–52). Here, stochastic processes and other environmental factors (e.g., vegetation) may play a much bigger role in structuring communities over time (in between years). The major shifts in AM fungal community composition within and between consecutive years in more benign ecosystems happen each spring, when the winter community supported by low photosynthetic carbon flux into roots is shifted to the summer community (with high photosynthetic carbon flux into roots) and the pattern of new community assembly during each year is largely stochastic (28). Yet this pattern may be less prominent in extreme environments, such as mofettes, and our results imply so.

Furthermore, the consistently high abundance of high $CO_2$ selected MOTU5 and MOTU22 in 2007, 2010, and 2011 together with smaller variability in diversity indices of high $CO_2$ AM fungal community (Fig. 1 and 4) suggests the stability of AM fungal community from high $CO_2$ sites of Stavešinci mofettes as a consequence of long-term abiotic selection, even when there is some seasonal variability. Although it is worth noting that dominant taxa within the control sites were also reasonably stable through time, however, there was greater variability in the composition of the rest of the community as we expected (H3). High $CO_2$ exposed AM fungal communities are dominated by a smaller number of competitive (or adapted) AM fungal taxa under permanent environmental selection. Importantly, in the context of the impacts of many human-derived environmental factors with a long-term nature on microbial communities, press-related changes and stability of microbial communities are severely under-investigated topics, and this study provides some novel insights.

## Stress tolerant AM fungal taxa

Despite the increased number of studies on drivers of AM fungal community ecology, characteristics of ecological niches of individual AM fungal taxa are still poorly understood and there is little information about organism-environment relationships among approximately species-level AM fungal taxa like AM fungal virtual taxa (VT) defined by MaarjAM database (17, 53). One possible use of long-term natural experiments (31) is

also in the study of AM fungal diversification that has recently been more thoroughly addressed based on the rapid growth of molecular data (54). Our results from high $CO_2$ exposed mofette sites show that the most abundant MOTUs compared to those from controls are MOTU5 and MOTU22, both assigned to Glomeraceae, where 90% and 96% sequences (for MOTU5, MOTU22, respectively) originated from high $CO_2$ locations. These two MOTUs are most similar to Glo85 and Glo86 from sampling in 2007 (29) and were consistently present in higher abundances in the high $CO_2$ exposed mofette locations in 2010 and 2011 (Fig. 3). Sampling conducted in 2007 showed that Glo85 and Glo86 were found exclusively in roots of several plant species growing at high $CO_2$ exposed locations in the Slovenian meadow site and were presented as potential high $CO_2$ specialist taxa (29). A closer look reveals that with higher resolution of sequencing (24,857 analyzed sequences) presented here, MOTU5 and MOTU22 are not entirely specific to the high $CO_2$ locations as was indicated by cloning and Sanger sequencing methods used by Maček et al. (29) (Fig. 3). As also shown with nestedness analyses (Fig. 1), mofettes have a clear environmental filtering caused by increased $CO_2$ leading to disappearance of sensitive species and increased abundance of $CO_2$ tolerant species. The dominant MOTU5 and MOTU22 present in high $CO_2$ exposed sites were both assigned to MaarjAM database virtual taxon VT309, putatively identified as *Rhizophagus* cluster [VT; following references (17, 53)]. VT309 is a widely distributed AM fungal taxon (source MaarjAM database), however it was firstly described in meadow mofette site in Slovenia (29) and later, following an intense increase in environmental studies of AM fungal communities in the last decade, it was found throughout several studies but interestingly, mainly in human-impacted environments (36–40, 55–57). Generally, those MOTUs increasing in abundance in high $CO_2$ exposed locations were present but exceptionally rare within control samples, suggesting that taxa that normally exist at the edge of their realized niche in the control site and are thus rare in benign environments may benefit from less competition in more extreme or stress-induced environments. Many soil organisms are locally rare, living, and surviving at the edge of their ecological niche. Where "patches" of suitable habitat occur, the abundance and dominance of these locally rare AM fungi may increase [e.g., references (27)], which may also be due to the competitive exclusion in these locations.

The high $CO_2$ exposed mofette sites, dominant VT309, were mostly reported from anthropogenically highly impacted habitats [35 accession numbers from anthropological ecosystem compared to 3 accession numbers from tropical moist broadleaf forest (58) and young pine forest (59), data from MaarjAM database; October 2023]. The presence of the virtual taxon VT309 (closest match to our MOTU5 and MOTU22) has been reported from vineyards (36, 37), heavy metal-contaminated soil (38), roots of maize grown in a 50-year monoculture (39), Italian rice fields (40), a rubber tree plantation in Northeast Thailand (55), and in a field and orchard treated with organic and inorganic fertilizer (56, 57). Maize monoculture, vineyards, and orchards are known for high concentrations of pesticide use (including copper contamination in soils as a result of long-term fungicide use in vineyards), and flooded rice fields are characterized by anaerobic conditions. The presence of VT309 in anthropogenically highly impacted habitats suggests that VT309 is a stress-tolerant AM fungal taxon. This, together with the highest abundance of MOTU5 and MOTU22, both assigned to VT309 across 2007, 2010, and 2011 (100%, 64%, and 90% MOTU5 sequences originated from high $CO_2$, and 100%, 97%, and 100% MOTU22 sequences originated from high $CO_2$, respectively) shows that this AM fungal taxon is the one capable of thriving in high $CO_2$ concentrations.

When comparing the most abundant MOTU in control sites within the years 2010 and 2011 (MOTU3 and MOTU31) with the study conducted in 2007 in the Stavešinci mofette site, we found that MOTU3 clusters with a very common and ubiquitous group of generalist AM fungal taxa *Rhizophagus irregularis*. Sequences from this cluster have been identified in control samples of the Stavešinci mofette site also in the 2007 sampling (identified as Glo23 in 29) but were not very abundant in 2007 and were exclusive to control areas. The second most abundant taxon in the control samples in

both sampling years (2010 and 2011) is MOTU31. It clustered together with Glo10 from the 2007 sampling and was present both in high $CO_2$ and control areas, however was relatively rare in the Stavešinci mofette area in 2007.

## Geographical distance context of mofettes

Similar patterns of AM fungal diversity have been observed in all the sampled mofette sites in Slovenia, Italy, and Czechia, with an exception of the forest mofette in Slovenia, supporting our hypothesis 5 for geographical context but not for different habitat types (grassland vs forest). Our models indicated that $CO_2$ did not have a statistically clear "global" effect on AM fungal diversity across study sites. However, for all three Hill measures of diversity (MOTU richness—$H_0$, MOTU evenness—$H_2$, and MOTU dominance—$H_\infty$), when we removed Slovenian forest sites from the data set, we once again found statistically clear effects of $CO_2$, suggesting potential interactions with habitat type (Fig. 4, Table 1).

Distinct AM fungal communities from different habitats (Slovenian forest vs meadow) and mofette locations (Slovenia, Italy, and Czechia) were found at different sites. Among the significant environmental vectors, the most variability in overall AM fungal sequence pool was explained by pH ($R^2 = 0.85$), followed by $CO_2$ concentration ($R^2 = 0.73$), total N ($R^2 = 0.70$), and concentration of $O_2$ ($R^2 = 0.65$) (Fig. 5; Fig. S3). This pattern is very similar to drivers of other soil microbial communities in mofette areas [e.g., reference (35)].

The AM fungal taxon, consistently most abundant in high $CO_2$ locations (MOTU5) in the Slovenian meadow mofette across sampling years has been abundant in high $CO_2$ locations also in all other mofette sites across Europe (Czech mofettes in Cheb Basin and the Italian mofette Bossoleto), while its sister taxon (MOTU22)—both are within the same VT309 reported from several anthropogenic, stressed environments [e.g., references (36–40, 55–57)]—has only been found in Stavešinci meadow mofette site with all the sequences originating from the meadow high $CO_2$ exposed locations. This could indicate that MOTU22 has evolved in the Stavešinci meadow mofette site and may be adapted. Further functional traits of these apparently stress adapted taxa would be interesting to study, however despite several attempts we have not yet managed to grow the high $CO_2$ adapted AM fungal taxa in culture and are, for now, known only as environmental sequences.

## Conclusion

The ecological response of AM fungal communities from areas with long-term soil hypoxia in several mofette sites in Europe (Slovenia, Italy, and Czechia) reflect a consistent community response to a long-term stress. This study demonstrates for the first time that long-term stress drives the assembly of a lower diversity subset of the local metacommunity that is significantly less variable in time than the surrounding community. Importantly, this study also reveals that in response to a specific stress (high $CO_2$), we can identify a number of taxa that are predictable members of these communities across wide geographic scales. The importance of long‐term experiments is stressed—in this case natural analogs allowing long-term studies—in understanding the impact of global change on terrestrial ecosystems (29, 31). Importantly, we demonstrate that the dynamics of AM fungal communities differ in response to long-term environmental stress with higher abundance of specialized taxa, that are typically rare in more benign environments. This result is part of an increasing body of evidence that the AM fungi display a wide range of definable life history and functional traits, suggesting that further research will reveal a predictive community ecology for the AM fungi. This is vital if we are to predict the effect of global change on AM-dominated ecosystems as changes in the composition of rhizosphere communities may impact aboveground plant communities and their productivity.

## MATERIALS AND METHODS

### Mofette sites

The study was conducted at four study sites: a meadow site (i) and a forest site (ii) located near Stavešinci in northeast Slovenia (*Slovenian meadow site* and *Slovenian forest site*, respectively), a Czech meadow site (iii) located in the Plesná valley in northwest Czechia (*Czech site*), and Italian meadow site (iv) located near Sienna in Tuscany (*Italian site*). The majority of the results come from the most intensively sampled *Slovenian meadow site*, while other mofette sites serve as supportive to better explain the patterns of AM fungal communities observed in mofettes. Each study site consists of a natural $CO_2$ spring (mofette area) with high geological $CO_2$ exposure and area without a $CO_2$ spring present (control area). Among these sites, the *Slovenian meadow site* has been sampled most throughout two consecutive years. The same locations in the *Slovenian meadow site* have been sampled to study AM fungal community ecology before [see reference (29)]; however, the new data and high-throughput sequencing technology presented in this paper add additional information to the patterns observed in the first study on this topic. *Italian* and *Czech*, as well as the *Slovenian forest site* mofette add additional spatial dimension to the *Slovenian meadow site* mofette data, however only single-time sampling has been conducted on the *forest* mofette and only single-time sampling in the *Italian* and *Czech* areas.

The *Slovenian meadow site* (46°36′49.2″N, 15°58′29.3″E) is a flat meadow with several mofette locations (60, 61), where ambient-temperature $CO_2$ of geological origin, with only traces of hydrogen sulphide (up to 0.0002%), methane (up to 0.2%), and carbon monoxide (up to 0.0001%) vents through the soil and into the atmosphere (35). The meadow mofettes have spatially and temporally well-described soil gas regime (32, 62) that has been stable for at least 30 years. The dominating vegetation at the study area is composed of *Agrostis canina* L., *Agrostis stolonifera* L., *Ajuga reptans* L., *Carex hirta* L., *Dactylis glomerata* L., *Holcus lanatus* L., *Juncus effusus* L., *Lathyrus pratensis* L., *Leontodon autumnalis* L., *Leucanthemum ircutianum* (Turcz.) DC., *Lotus corniculatus* L., *Lythrum salicaria* L., *Plantago lanceolata* L., *Poa pratensis* L., *Polygonum aviculare* L., *Scirpus sylvaticus* L., *Taraxacum officinale* Weber and Wiggers, *Trifolium pratense* L., and *Veronica officinalis* L. (detailed botanical survey in Table S1). Soils from high $CO_2$ exposed areas have a lower pH and a higher content of total nitrogen and available phosphorus compared to control samples [Table S2; reference (31)]. To examine the intra- and inter-annual stability of AM fungal community from Slovenian meadow, we have also used the meteorological data (Table S7) from Gornja Radgona weather station, located within 8 km of Slovenian meadow sites. The data were obtained from the Slovenian Environmental Agency (ARSO; http://www.arso.gov.si). Mean minimum and maximum temperature, duration of sunlight in hours, and total amount of rain as environmental factors were correlated with AM fungal community data. These climate variables were calculated for each of the eight time points for 30-day period, finishing on the last day of sampling. A summary of the climate variables calculated for use in this study is included in Supplemental material (Table S7).

The *Slovenian forest site* (46°37′27.7″N, 15°59′12.1″E) has two larger vents filled with water (63), where ambient-temperature $CO_2$ of geological origin vents through the soil carrying traces of hydrogen sulphide (up to 0.0004%), methane (up to 0.3%), and carbon monoxide (up to 0.0003%) into the atmosphere. The vegetation in the forest consists of *Picea abies* (L.) Karsten, *Caprinus betulus* L., *Acer pseudoplatanus* L., *Quercus robur* L., *Pinus sylvestris* L., *Frangula alnus* Mill., *Fagus sylvatica* L., *Corylus avellana* L., *Castanea sativa* L., *Carex brizoides* L., and *Maianthemum bifolium* (L.) F. W. Schmidt (64). Soils from high $CO_2$ exposed areas have a higher content of total nitrogen and available phosphorus compared to control samples (Table S3).

The *Czech site* is a flat meadow in the flood plain of the Plesná River (12°27′E, 50°07′N), Czechia, where the migrating gas consists of up to 99% $CO_2$ and can contain traces of hydrogen, helium, argon, methane, oxygen, and nitrogen (65). The site is known for

its long-term stability, since it was probably active since the late Pleistocene (66, 67). The vegetation consists of perennial, non-woody herbs and grasses with dominating *Achillea millefolium* L., *Alopecurus pratensis* L., *Carex brizoides* L., *Carex nigra* (L.) Reichard, *Deschampsia caespitosa* (L.) PB., *Festuca rubra* L., *Filipendula ulmaria* (L.) Maxim., *Holcus lanatus* L., *Lotus pedunculatus* Cav., and *Polygonum bistorta* L (68). Lower pH values and a higher content of total nitrogen and available phosphorus in soils from high $CO_2$ exposed area compared to control samples have been measured for this area (Table S4).

The *Italian site* Bossoleto (43°17′N, 11°35′E) is shaped as a round shaped sinkhole of about 80 m in diameter and 20 m in depth. In this sinkhole, originally described as early as 1899 (69), a $CO_2$ lake forms every night due to the combination of the site topography and $CO_2$ accumulation where $CO_2$ concentrations range from 0.04% to 80% in 24-h period (47). The detailed chemical and isotopic composition of the soil gasses from the Italian mofettes is described in Fazi et al. (47). Briefly, soil gas at the mofette area can contain up to 190 mmol mol$^{-1}$ N$_2$, 4.9 mmol mol$^{-1}$ Ar, 0.18 mmol mol$^{-1}$ H$_2$S, 41 mmol mol$^{-1}$ CH$_4$, and 4.5 mmol mol$^{-1}$ C$_2$H$_6$ (47). The vegetation is described in Miglietta et al. (70) and consists mostly of *Agrostis stolonifera* L., *Plantago lanceolata* L., *Quercus ilex* L., and *Quercus pubescens* Willd. At the bottom of the valley, there is the dominance of *Agrostis stolonifera* L. and *Phragmitas australis* (Cav.) (47). Lower pH and higher available phosphorus values in soils from the high $CO_2$ exposed areas compared to control areas were measured (Table S5).

## Soil and plant root sampling

### *Soil gas measurements and analyses*

To quantify soil $CO_2$ levels from Slovenian meadow, Slovenian forest, and Czech sites, soil gas composition was measured with a portable gas analyzer (GA2000+, Ansyco, Germany) at 15 cm depth in the middle of each sampled soil core. The gas analyser measures carbon dioxide ($CO_2$) and methane simultaneously by infrared absorption and oxygen, carbon monoxide, and hydrogen sulphide by internal electrochemical cell measurements (62, 71). At Italian sites, soil $CO_2$ efflux was measured with LI-6400-09 Soil $CO_2$ Flux Chamber (LI-COR, Lincoln, USA). Soil chemical analyses were performed on soil taken from sampled soil cores at each time of sampling from all four sampled study sites. Soil pH was measured in a suspension of soil in 0.01 M CaCl$_2$ (72). Total nitrogen (N) content was determined after dry combustion (73), and easily extractable phosphorus (P$_2$O$_5$) in AL solution (74) was determined colorimetrically (75).

### *Plant root sampling for molecular analyses*

*Slovenian meadow site*: Soil cores were collected from three spatially independent mofette vents within the Slovenian meadow site, each separated by at least 14 m to ensure spatially independent sampling of AM fungi. From each of these three locations, a mofette area with an average of 71% $CO_2$ (high $CO_2$ exposed samples; *n* = 96) in soil air, and a control area with <2.9% $CO_2$ (control samples; *n* = 96) in soil air, were sampled at eight-time points: April 2010, June 2010, August 2010, October 2010, April 2011, June 2011, August 2011, and October 2011. Soil $CO_2$ and $O_2$ concentrations were confirmed immediately before sampling for each sampling core (Table S2). Four replicate soil cores (pseudo-replicates; 10 cm wide, 15 cm deep) were taken from each mofette and control area at each time point. Plant species growing above each collected soil core were identified (Table S1) and plant height was measured (Table S6). Soil cores were washed to remove soil particles and roots were collected. Prior to downstream DNA processing, a subsample of ≈60% of all roots collected in 2010 was dried at 40°C and stored at room temperature, whereas roots collected in 2011 were first stored at −20°C and then dried at 40°C and stored at room temperature.

*Slovenian forest site:* Soil cores were collected from a mofette area with more than 58% $CO_2$ (high $CO_2$ exposed samples; *n* = 32) in soil air and a control area with <1.6% $CO_2$ (control samples; *n* = 32) in soil air. Soil cores were sampled at eight-time points and processed as described for Slovenian meadow samples.

*Czech site*: Roots of *Deschampsia caespitosa* (L.) PB. were sampled in May 2008 from a mofette area (high $CO_2$ exposed samples; *n* = 7) with 78.7 ± 20.7% $CO_2$ measured in soil air and from control area (control samples, *n* = 6) with 0.8 ± 0.6% $CO_2$ in soil air. Roots from individual plants were washed and a subsample of ≈60% of all roots collected was dried at 40°C and stored at room temperature.

*Italian site*: Roots of *Agrostis stolonifera* L., *Plantago lanceolata* L., and *Centaurea alba* L. were collected from a mofette area (high $CO_2$ exposed samples; *n* = 3 for *A. stolonifera*, *n* = 4 for *P. lanceolata*, and *n* = 3 for *C. alba*) with in average >170 µmol $CO_2$ $m^{-2}$ $s^{-1}$, corresponding to *ca* 60% $CO_2$ in soil air (based on positive correlation of $CO_2$ flux and $CO_2$ concentrations in soil; 32) and from a control area with in average <22 µmol $CO_2$ $m^{-2}$ $s^{-1}$, corresponding to *ca* 2% $CO_2$ in soil air (control samples *n* = 4 for *A. stolonifera*, *n* = 3 for *P. lanceolata*, and *n* = 3 for *C. alba*) in June 2008. Plants were selected based on measured $CO_2$ flux from soil with LI-6400 (LI-COR, Lincoln, USA). Roots from individual plants were washed and a subsample of ≈60% of all roots collected was dried at 40°C and stored at room temperature.

## Molecular methods

In order to examine the AM fungal community from the plant roots from each sample, we used 454 GS-FLX pyrosequencing of amplicons of the 18S small subunit (SSU) region of ribosomal RNA (rRNA). Plant roots for each individual sample were first homogenized in a Retsch mixer mill (Retsch, Germany) and DNA extracted from 50 mg of homogenate using MoBio PowerPlant (2010 samples from Slovenian meadow, Slovenian forest, Czech, and Italian sites) or PowerSoil DNA isolation kits (2011 samples from Slovenian meadow and Slovenian forest sites) following the manufacturer's instructions (MoBio Laboratories, Inc., Carlsban, CA, USA).

To produce amplicon libraries for 454 pyrosequencing, a 550-bp partial fragment of the SSU rRNA gene was first amplified by PCR using hot start Taq DNA polymerase (Promega Co., Madison, WI, USA or Qiagen Ltd. Crawley, UK), the universal eukaryotic primer NS31 (76) and the primer AM1, which excludes plants and amplifies the major glomeromycotan families (14). Moreover, this primer-pair provides accurate repeatability and no detectable PCR-biases (18). PCR was carried out in a 50-µL reaction volume with 1 µL of DNA template, 2 mM dNTPs, 10 pmol of each primer (PCR conditions: 95°C for 2 min (Promega Taq) or 5 min (Qiagen Taq); 35 cycles at 94°C for 30 s, 64°C for 40 s, and 70°C for 1 min; and 70°C for 5 min) on a Techne TC-512 (Techne Co, Stone, Staffs, UK). PCR products were purified using QIAquick PCR Purification Kit (Qiagen Ltd, Crawley, UK). Based on similar concentrations of DNA determined using the NanoDrop 1000 Spectrophotometer (Thermo Scientific, Wilmington, DE, USA), pseudo-replicate samples were pooled. A secondary semi-nested PCR was then used to add the fusion primers required for 454 sequencing, which contained the GS-FLX adaptors A and B, a multiplex identifier (MID), and a forward primer WANDA (28). The secondary PCR was performed using hot start Taq DNA polymerase (Promega Co., Madison, WI, USA or Qiagen Ltd. Crawley, UK) and carried out in a 50-µL reaction volume with 1 µL of PCR template (of pooled replicate samples), 2 mM dNTPs, 10 pmol of each primer (PCR conditions: 95°C for 2 min; 10 cycles at 94°C for 0.5 min, 60°C for 0.5 min, and 72°C for 1 min; and 72°C for 10 min) on a Techne TC-512 (Techne Co, Stone, Staffs, UK). The PCR products were purified using QIAquick Gel Extraction Kit (Qiagen), and then quantified using qPCR. Equimolar concentrations of 24 MID-tagged samples were loaded into individual lanes on GS-FLX Titanium plates separated with an eight-lane gasket (454 Life Sciences ∕ Roche Applied Biosystems, Nutley, NJ, USA) and sequenced at the Food and Environment Research Agency (FERA), UK.

## Bioinformatics

The QIIME pipeline (77) was used for all bioinformatics analysis of the 454 data and to define molecular operational taxonomic units (MOTUs), following methods detailed in Dumbrell et al. (19). Sequences shorter than 400 bp and longer than 650 bp, sequences

with an average quality score below 25 and sequences without barcode sequences or without the correct primer sequences were removed. Raw 454 data were denoised using the flowgram denoiser algorithm (78) and chimeras were removed via the *de novo* chimera checker associated with USEARCH (79). USEARCH (79) clustering at 97% sequence similarity threshold was used to define MOTUs. MOTUs with fewer than two sequences were removed. All remaining MOTUs were compared with their closest phylogenetic relatives from GenBank using the BLASTn algorithm (80) and MOTUs of non-Glomeromycotan origin were removed from the data set. We also removed samples containing fewer than 229 sequences in total. To account for differences in the sequencing depth per sample, we normalized the data using rarefaction [Rarefy in R package GUniFrac; reference (81)] which is a statistically and ecologically valid normalization strategy (82), but also checked for qualitative differences to the results using an alternate form of normalisation, DESeq2 (83), the results for which are shown in the Supplemental material (Figs. S5 to S9).

## Data analysis

Representative sequences from each AM fungal MOTU were compared to the Maarj*AM* database (17) to determine their closest matched virtual taxa ([VT; reference (17)]. The closets matched VT and representative sequences from Maček et al. (29) were aligned with representative sequences from each MOTU (Fig. S1) using ClustalW (84). A phylogenetic consensus tree was built Jukes-Cantor (85) substitution model, Neighbor-Joining phylogeny (86), and *Corallochytrium limacisporum*, a choanozoan, as a general outgroup to all fungi (87). Phylogenetic support was calculated via bootstrapping with 10,000 pseudo-replicates (88). All phylogenetic analyses were performed using Geneious version 5.5.7 (89).

In order to test whether a distinct and consistent community of AM fungi exists at high $CO_2$ exposed samples from the Slovenian meadow sites, the differences in AM fungal α-diversity and community composition between high $CO_2$ exposed and control areas were quantified by Hill numbers (90) of order 0 ($H_0$), 2 ($H_2$), and infinity ($H_\infty$), which represent species richness, community evenness (inverse Simpsons index) and community dominance respectively (inverse Berger-Parker index). Differences in species richness ($H_0$) were tested for using a negative binomial generalized linear mixed effects model, with the sampling period specified as a random effect. Differences in the other two α-diversity metrics were tested using linear mixed-effects models, again specifying the sampling period as a random effect.

Differences in within-, and between-, group community dissimilarity were tested using the PERMANOVA test implemented in the vegan package (see adonis function), with 1,000 permutations, and visualized using non-metric multidimensional scaling (NMDS) analysis. Pairwise community dissimilarity was quantified using the Sorensen index. We also decomposed the Sorensen dissimilarity matrix into species turnover and nestedness components, as described by Baselga (91), and performed further PERMANOVA analyses to test whether any differences were due to nestedness effects or species turnover. After sorting the MOTU table rows and columns by decreasing MOTU occupancy and richness, respectively, the community nestedness was further examined by calculating matrix temperature. Matrix temperature quantifies the extent to which a community presence absence matrix follows a perfect nestedness isocline, where 0 indicates perfect nestedness and 100 shows no nestedness. As the process of shuffling the presence-absence matrix can generate different results each time, we repeated the process 1,000 times and reported mean matrix temperature and standard error.

The obtained differences between high $CO_2$ exposed and control AM fungal community were also visualised using CCA ordinations with fitted significant ($P < 0.05$ based on 10,000 permutations) environmental vectors onto the CCA ordination. The environmental vectors used in the analysis were: soil factors [soil $CO_2$ and $O_2$ concentrations, pH, total nitrogen (N), and available phosphorus ($P_2O_5$)]; climate variables

(maximal and minimal temperature, sun hours, and total rainfall in the 30 days prior to sampling) and seasonality (year and month) (Table S2 and S7).

The stability of AM fungal communities at Slovenian meadow sites across seasons and over 4 years (2007–2011) and yearly fluctuations in the dominant taxon's identity were studied by examining the MOTU proportional abundance distribution of the AM fungal community. These are displayed as rank-abundance plots, and by proportional abundance plots of two of the most abundant MOTUs from mofette and control areas within Slovenian meadow sites for each sampling time from June to August 2007 (29) and from April 2010 to October 2011.

To determine whether the dominant taxa in control and mofette habitats experience temporal shifts, we used multivariate negative binomial generalized linear models (MV-GLMs) on rarefied MOTU data. For each MOTU, we ran two GLMs, one with only $CO_2$ level (high $CO_2$ exposed vs control), and one with $CO_2$ level and sampling period. We determined whether there were statistically clear temporal shifts in abundance using likelihood ratio tests to test between these two models. Adjusted univariate $P$ values were calculated by 1,000 Monte-Carlo resampling permutations. Under our hypothesis, we predict that MOTUs that are more abundant in high $CO_2$ exposed mofette habitats will not show statistically clear temporal shifts in abundance, whereas those that are more abundant in control $CO_2$ conditions will show statistically clear temporal shifts.

Finally, to examine whether the influence of abiotic extremes is independent of habitat and geographical context we quantified the diversity of the AM fungal communities in each of the study sites described above (including the Slovenian meadow site focussed on by the previous three hypotheses). Diversity was quantified as Hill numbers (90) of order 0, 2, and infinity, as described for hypothesis 1. Statistical differences were tested for using a random intercept and slope GLMM ($H_0$—MOTU richness) or LMMs ($H_2$ and $H\infty$, evenness and dominance, respectively), where both the intercept and $CO_2$ coefficient are conditional upon site.

The differences in overall AM fungal community composition across all four study sites were also visualised using CCA ordinations with fitted significant ($P < 0.05$ based on 10,000 permutations) environmental vectors, i.e., soil $CO_2$ and $O_2$, total nitrogen, available phosphorus onto the CCA ordination (Table S2 through S5 and S7). To investigate how the abundance of AM fungal MOTUs changed among mofette and control areas across all four studied sites, (MV-GLMs were used on rarefied data sets (229 sequences per sample). Both $CO_2$, site, and their interaction were entered as model terms.

Three-way ANOVA was used to examine the differences in soil characteristics among different gas regimes, sample months, and years. Non-parametric Kruskal-Wallis rank-sum test was used to examine the differences in the height of plants between high $CO_2$ sites and control sites. Spearman's rank correlation was used to correlate the concentrations of measured soil $CO_2$ with measured soil pH values. All statistical, diversity, and community analyses were conducted on normalized number of sequences using R statistical language version 3.5.0 (92) with standard R libraries; the community analysis specific package "vegan" (93), "mvabund" for analysing multivariate abundance data (94), "GUniFrac" (81) for normalizing data, and "Rcmdr" R commander for basic statistics (95).

## ACKNOWLEDGMENTS

The authors would like to thank Hardy Pfanz and Antonio Raschi for enabling access to the mofette sites in Italy and Czechia, Miha Šijanec, and Anita Dirnbek for the help with sampling and sample preparation at Slovenian Stavešinci mofettes, Urška Videmšek for help with sampling at Czech and Italian mofettes, and Boris Turk and Klemen Eler for botanical survey conducted at Stavešinci meadow mofettes.

This work was supported by the Slovenian Research Agency (ARRS) projects J4-5526 and J4-7052, awarded to I.M., a Royal Society International Joint Project awarded to I.M. and T.H., Slovenian Research Agency (ARRS) programme P4-0085 and P4-0107.

The authors declare that there are no conflicts of interest. I.M., A.J.D., and T.H. designed the research. I.M. and N.Š. performed the research (field and laboratory work). A.J.D., D.R.C., and N.Š. analyzed data. I.M., A.J.D., D.R.C., N.Š., and T.H. wrote the paper.

## AUTHOR AFFILIATIONS

[1]Department of Agronomy, Biotechnical Faculty, University of Ljubljana, Ljubljana, Slovenia

[2]Department of forest physiology and genetics, Slovenian Forestry Institute, Ljubljana, Slovenia

[3]School of Life Sciences, University of Essex, Colchester, United Kingdom

[4]Institute for Analytics and Data Science, University of Essex, Colchester, United Kingdom

[5]Department of Biology, University of York, York, United Kingdom

[6]Institute for Ecology and Evolution, School of Biological Sciences, University of Edinburgh, Edinburgh, Scotland

[7]Department of Biology, Biotechnical Faculty, University of Ljubljana, Ljubljana, Slovenia

## PRESENT ADDRESS

Nataša Šibanc, Slovenian Forestry Institute, Ljubljana, Slovenia

Thorunn Helgason, Institute for Ecology and Evolution, School of Biological Sciences, University of Edinburgh, Edinburgh, Scotland, United Kingdom

## AUTHOR ORCIDs

Nataša Šibanc  http://orcid.org/0000-0001-8979-9450
Dave R. Clark  http://orcid.org/0000-0002-2306-0220
Alex J. Dumbrell  http://orcid.org/0000-0001-6282-3043
Irena Maček  http://orcid.org/0000-0002-5945-5582

## FUNDING

| Funder | Grant(s) | Author(s) |
| --- | --- | --- |
| Javna Agencija za Raziskovalno Dejavnost RS (ARRS) | J4-5526, J4-7052, P4-0085 | Irena Maček |
| Javna Agencija za Raziskovalno Dejavnost RS (ARRS) | P4-0107 | Nataša Šibanc |

## AUTHOR CONTRIBUTIONS

Nataša Šibanc, Data curation, Formal analysis, Investigation, Visualization, Writing – original draft, Writing – review and editing | Thorunn Helgason, Conceptualization, Data curation, Formal analysis, Investigation, Methodology, Software, Supervision, Validation, Visualization, Writing – original draft, Writing – review and editing | Alex J. Dumbrell, Conceptualization, Data curation, Formal analysis, Investigation, Methodology, Software, Supervision, Validation, Visualization, Writing – original draft, Writing – review and editing | Irena Maček, Conceptualization, Data curation, Formal analysis, Funding acquisition, Investigation, Methodology, Project administration, Resources, Software, Supervision, Validation, Visualization, Writing – original draft, Writing – review and editing.

## DATA AVAILABILITY

All data and R code required to reproduce our statistical analyses have been deposited into a Figshare collection. OTU centroid sequences have also been deposited into genbank (accession numbers from LK937169 to LK937200 and from MG461856 to MG461860).

## ADDITIONAL FILES

The following material is available online.

### Supplemental Material

**Supplemental Material (mSystems01331-23-s0001.docx).** Supplemental tables and figures.

### Open Peer Review

**PEER REVIEW HISTORY (review-history.pdf).** An accounting of the reviewer comments and feedback.

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
