## [Reviewer comments · mSystems]

Extreme Environments Simplify Reassembly of Communities of Arbuscular Mycorrhizal Fungi

Nataša Šibanc, Dave Clark, Thorunn Helgason, Alex Dumbrell, and Irena Macek

Corresponding Author(s): Irena Macek, Univerza v Ljubljani Biotehniška fakulteta

Review Timeline:

Submission Date:

December 13, 2023

Accepted:

January 18, 2024

Editor: Michaeline Albright

Reviewer(s): The reviewers have opted to remain anonymous.

Transaction Report:

DOI: <https://doi.org/10.1128/mSystems.01331-23>

Re: mSystems01331-23 (Extreme Environments Simplify Reassembly of Communities of Arbuscular Mycorrhizal Fungi)

Dear Dr. Irena Macek:

Thank you for your thorough responses to comments in previous reviews. Your manuscript has been accepted, and I am forwarding it to the ASM production staff for publication. Your paper will first be checked to make sure all elements meet the technical requirements. ASM staff will contact you if anything needs to be revised before copyediting and production can begin. Otherwise, you will be notified when your proofs are ready to be viewed.

Featured Image Submissions: If you would like to submit a potential Featured Image, please email a file and a short legend to msystems@asmusa.org. Please note that we can only consider images that (i) the authors created or own and (ii) have not been previously published. By submitting, you agree that the image can be used under the same terms as the published article. Image File requirements: TIF/EPS, 7.5 inches wide by 8.25 inches tall (at least 2,250 pixels wide by 2,475 pixels tall), minimum 300 dpi resolution (600 dpi preferred), RGB, and no figure elements, e.g., arrows or panel labels. The legend should be a short description of the image, 1-2 sentences recommended.

Sincerely,
Michaeline Albright
Editor